# Comparison of Different Analytical Strategies for Classifying Invasive Wetland Vegetation in Imagery from Unpiloted Aerial Systems (UAS)

**Louis Will Jochems** [1,*], **Jodi Brandt** [1], **Andrew Monks** [2], **Megan Cattau** [1], **Nicholas Kolarik** [1], **Jason Tallant** [3] **and Shane Lishawa** [2]

1   Human-Environment Systems Research Center, Boise State University, 1910 University Dr., Boise, ID 83725, USA; jodibrandt@boisestate.edu (J.B.); megancattau@boisestate.edu (M.C.); nicholaskolarik@u.boisestate.edu (N.K.)
2   Institute of Environmental Sustainability, Loyola University Chicago, 1032 W. Sheridan Rd., Chicago, IL 60660, USA; amonks@luc.edu (A.M.); slishawa@luc.edu (S.L.)
3   Department of Biological Sciences, University of Michigan Biological Station, 9133 Biological Rd., Pellston, MI 49769, USA; jtallant@umich.edu
*   Correspondence: louisjochems@u.boisestate.edu

**Abstract:** Detecting newly established invasive plants is key to prevent further spread. Traditional field surveys are challenging and often insufficient to identify the presence and extent of invasions. This is particularly true for wetland ecosystems because of difficult access, and because floating and submergent plants may go undetected in the understory of emergent plants. Unpiloted aerial systems (UAS) have the potential to revolutionize how we monitor invasive vegetation in wetlands, but key components of the data collection and analysis workflow have not been defined. In this study, we conducted a rigorous comparison of different methodologies for mapping invasive Emergent (*Typha* × *glauca* (cattail)), Floating (*Hydrocharis morsus-ranae* (European frogbit)), and Submergent species (*Chara* spp. and *Elodea canadensis)* using the machine learning classifier, *random forest*, in a Great Lakes wetland. We compared accuracies using (a) different spatial resolutions (11 cm pixels vs. 3 cm pixels), (b) two classification approaches (pixel- vs. object-based), and (c) including structural measurements (e.g., surface/canopy height models and rugosity as textural metrics). Surprisingly, the coarser resolution (11 cm) data yielded the highest overall accuracy (OA) of 81.4%, 2.5% higher than the best performing model of the finer (3 cm) resolution data. Similarly, the Mean Area Under the Receiving Operations Characteristics Curve (AUROC) and F1 Score from the 11 cm data yielded 15.2%, and 6.5% higher scores, respectively, than those in the 3 cm data. At each spatial resolution, the top performing models were from pixel-based approaches and included surface model data over those with canopy height or multispectral data alone. Overall, high-resolution maps generated from UAS classifications will enable early detection and control of invasive plants. Our workflow is likely applicable to other wetland ecosystems threatened by invasive plants throughout the globe.

**Keywords:** data fusion; classification algorithms; classification approach; distribution mapping; early detection; invasion monitoring; invasive species; multiscale assessment; spatial resolutions

## 1. Introduction

Wetlands are important ecosystems facing many threats across the globe [1]. Invasive plants are one of the greatest threats to wetlands, as they can reduce native biodiversity, habitat quality, and oxygen levels in the water column [2,3]. One effective way to mitigate plant invasions is to identify them early in the invasion process through early detection and rapid response protocols (EDRR) [4]. An emerging component of EDRR is the use of unpiloted aerial systems (UAS) to complement field-based monitoring [4]. UAS have great potential to classify, map, and detect new stands of invasive plants [5–7]. However,

effective workflows have not yet been established for flying UAS over wetlands and efficiently detecting invasive plants in UAS imagery.

There are various aspects of the UAS workflow that are relatively well established in terrestrial systems but less so in wetland systems. First, there is a trade-off between spatial resolution and image extent. UAS produces imagery with high spatial resolution (minimum resolvable object sizes in cm) that can be used to map individual plant species of smaller growth forms, such as herbaceous plants [5,6,8,9] or grasses [10,11]. Specifically, the centimeter-scale ground sampling distance (GSD, or minimum pixel size) can capture unique characteristics of invasive species, such as flowering umbels [6], thereby distinguishing the target species from the surrounding plant community that would otherwise be obscured within pixels of coarser resolution imagery [5–7,12]. The main disadvantage of flying UAS at altitudes necessary to detect small invasive plants (<5 cm pixels) is that the maximum spatial extent of UAS imagery may be limited in capturing widespread invasions [6,7,13,14] by flight duration (and flight safety) and increased processing time due to the additional data capture. Since the user controls the altitude at which to fly UAS (following local flight laws), s/he can compare different altitudes and determine the spatial resolution that best captures important characteristics of smaller invasive plants while covering extents that are relevant for EDRR purposes. The trade-off between spatial resolution and image extent has not been well studied in wetlands.

A second understudied component of invasive plant detection with UAS in wetlands is the comparison of pixel-versus object-based methods for image classification [15,16]. While centimeter-scale pixels seem ideal for distinguishing subtle spectral and structural differences between vegetation types, the features of interest, or image objects, are often larger than the spatial resolution and/or GSD of the UAS imagery (e.g., stands of invasive vegetation). Pixel-based approaches involve classifying individual pixels throughout the image, while object-based approaches involve a preliminary step of segmenting pixels into "objects" based on similar spectral and structural values, prior to classification [15,17]. Most off-the-shelf UAS sensors provide high spatial resolution but low spectral/radiometric resolution, so additional information (e.g., spatial patterns and context) through object-based approaches may be necessary [9]. Previous work has found object-based methods to be more effective at accurately mapping invasive plant species in UAS imagery of different ecosystems [5,6,18]. Since invasive plant species form monotypic stands at various densities within wetlands [6,19], object-based segmentation algorithms may be necessary to group pixels into ecologically appropriate objects that provide additional geometric attributes (e.g., edge-to-area ratio, minimum/maximum segment size, etc.) for image classification of a single-date image [20]. However, pixel-based approaches may be preferable when the minimum mapping unit, e.g., an individual of a small plant species, may be equal to or smaller in size than the GSD (pixel sizes) in spatially heterogeneous environments. Lastly, pixel-based approaches do not require the computationally intensive segmentation process, and thus can deliver critical information to end-users (managers) more quickly than object-based approaches. It is unclear which method (pixel- vs. object-based) and data (i.e., imagery with different spatial resolutions) will yield more accurate maps of invasive wetland vegetation in UAS imagery.

The inclusion of structural measures from UAS flights for invasive plants may enable more accurate detection, but merits further study in wetlands. A single flight mission provides remote sensing data that can characterize differences in spectral reflectance, structure (e.g., canopy height and growth form), and/or texture (i.e., local spatial variability in a moving window of pixels) [21] between species in target communities [7,22,23]. It can be difficult to differentiate plant species across functional groups using multispectral sensors alone [24,25], but canopy height (in m) derived from Structure from Motion (SfM) in UAS flights can supplement machine learning classifiers in mapping vegetation [7,8,11,26,27]. Accurate 3D reconstruction of vegetation structure requires adequate distribution of ground control points (GCPs) for georeferencing features in the flight footprint, as well as adjusting forward and side overlap of individual images captured by the UAS sensor [28–30]. The in-

corporation of accurate structural (or textural) metrics is likely necessary for differentiating spatially mixed vegetation classes with similar spectral signatures during peak growing season [6,7,12,16,20,31]. Therefore, leveraging structural data from UAS to map invasive vegetation in wetlands where many species of various growth forms (e.g., floating and emergent plants) often grow in close proximity to one another [7,32] shows great promise for EDRR purposes.

The aim of this study was to explore the effectiveness of UAS data and machine learning algorithms in mapping the invasive species, floating *Hydrocharis morsus-ranae* (European frogbit; hereafter EFB) and emergent *Typha × glauca* (cattail), along with submergent species and other vegetation (native and non-native) in a wetland near Lake Huron, MI, USA. EFB is a small free-floating aquatic plant (individual leaves at 5–6 cm in diameter) that has established in a limited number of wetlands in Michigan, and often grows within or around taller invasive cattails in Great Lakes wetlands [32]. In contrast, the emergent leaves of invasive cattails can grow up to 2 m tall at full maturity, forming dense stands in wetlands throughout the region [32–34]. Thus, the invasive stage of EFB and dominant stands of cattails provided an ideal study system for creating an UAS workflow for EDRR of invasive species at various growth forms. We collected UAS imagery during the peak growing season (August 5th) and then sought to achieve the following objectives: compare classification accuracies of invasive-dominant vegetation classes between datasets with (a) different spatial resolutions (3 cm vs. 11 cm pixels), (b) via two common classification approaches (pixel- vs. object-based), and (c) multispectral only vs. multispectral plus structural variables at coarser spatial resolutions.

## 2. Materials and Methods

### 2.1. UAS, Study Site, & Data Collection

Our research team collected multispectral UAS imagery flown over a wetland in Alpena Wildlife Refuge, Alpena, Michigan, USA during the peak growing season (5 August) in 2019 (Figure 1). We used a Parrot Bluegrass Quadcopter™ equipped with a Parrot Sequoia Sensor™ (Paris, France) with Green (530–570 nm), Red (640–680 nm), Red Edge (730–740 nm), and Near Infrared (NIR; 770–810 nm) bands. We flew at ~105 m altitude to capture high-resolution imagery (~11 cm pixels) over 10.1 ha (6.3 km$^2$) and at ~30 m altitude to capture very-high resolution imagery (~3 cm) over 4.73 ha (4.3 km$^2$). We flew all flights in a grid pattern with 85% front and side overlaps due to the complex heterogeneity of vegetation and flat terrain throughout the site (with the exception of the tree canopy; Figure 1), both of which require higher image overlaps to extract adequate matching points for 3D point clouds [35]. We parameterized all flight missions in the Pix4Dcapture app on a field tablet.

The flight footprints covered plant communities primarily consisting of dominant/invasive stands of emergent *Typha × glauca* (cattails) and floating EFB. There were also interspersed abundances of floating *Lemna minor* or *Spirodela polyrhiza* (duckweeds) and larger lily species, primarily *Nymphaea odorata*, throughout the various wetland functional types or zones of interest (Figures 1 and 2). Less dominant species included emergent *Schoenplectus* spp. (bulrush) and submergent vegetation (*Chara* spp., *Ceratophyllum demersum*, *Elodea canadensis*, *Myriophyllum* spp., and *Potamogeton crispus*). Other features included terrestrial vegetation (mixed evergreen-deciduous forest) on an island, open water (areas with at least 1 m of unvegetated water depth), and *Typha × glauca* leaf litter. On the flight date, most of the aforementioned species were fully mature throughout the site.

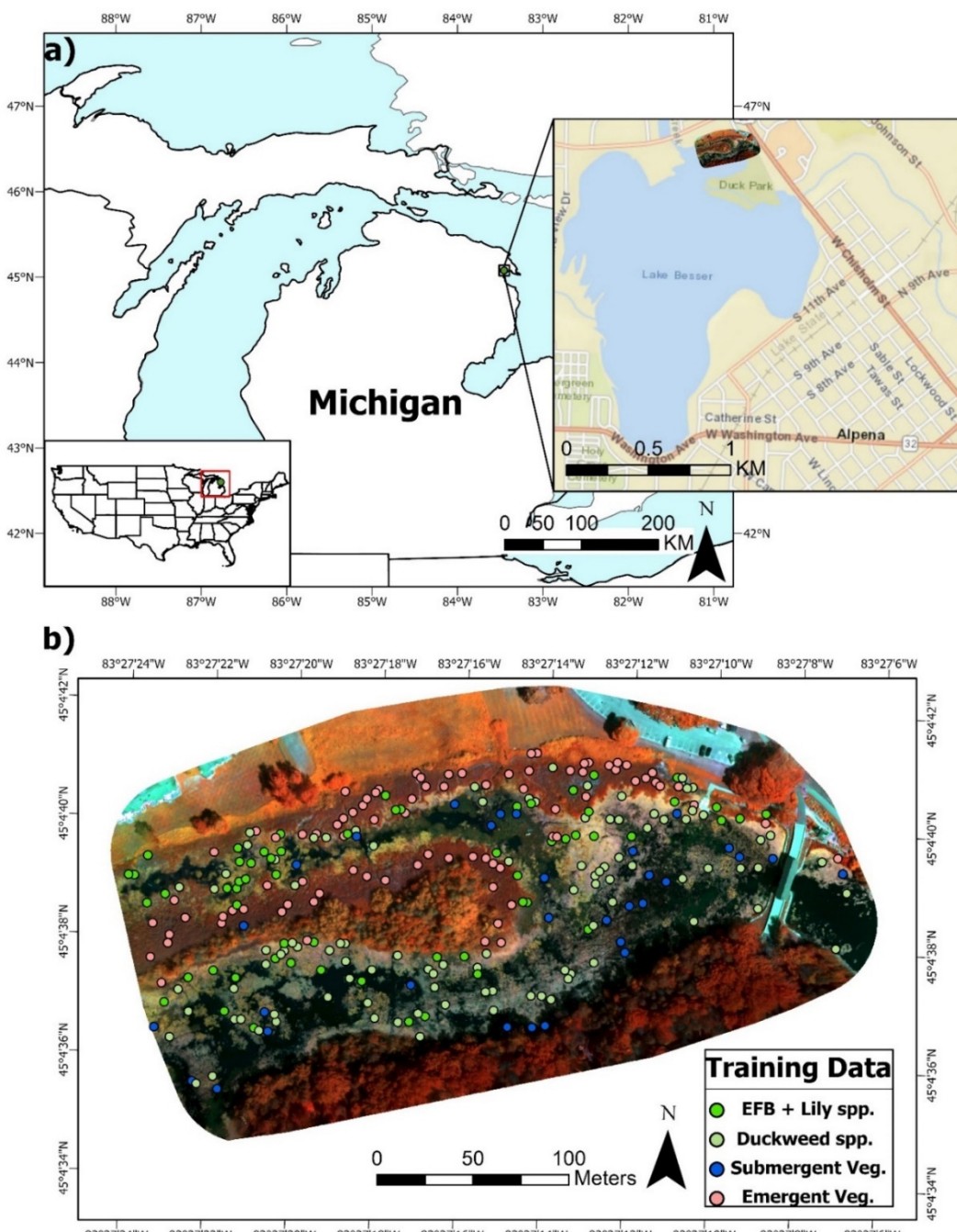

**Figure 1.** (**a**). Location of the study site and flight footprint in Alpena, MI, USA (approximate centroid (dd): 45.077499 °N, 83.454249 °W). (**b**). Multispectral false color composite (Red: NIR, Green: Green, Blue: Red Edge) of the 11 cm dataset with overlaid ground reference points of the four primary wetland vegetation classes for training the RF algorithm.

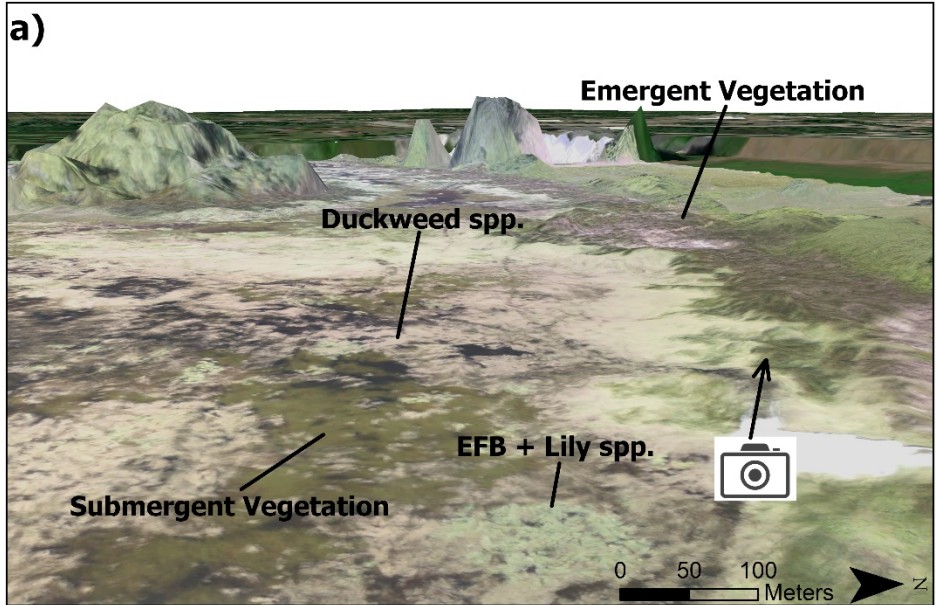

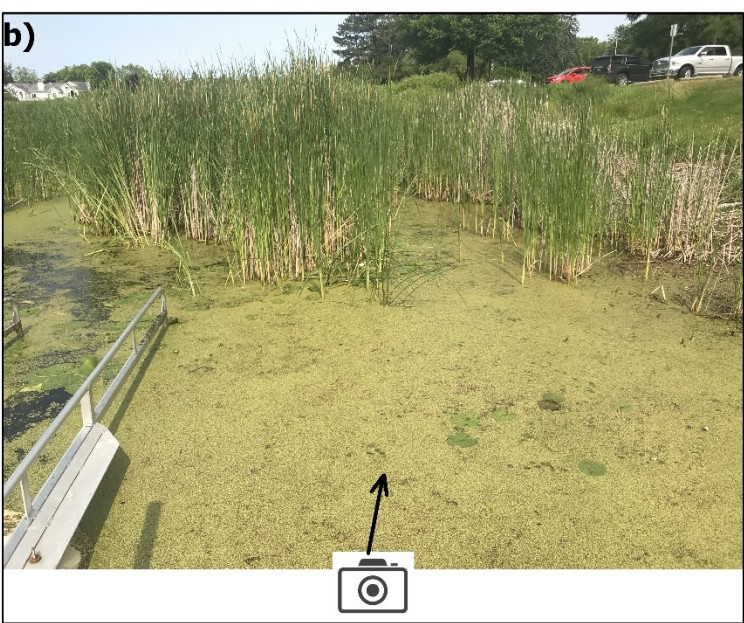

**Figure 2.** (**a**). The RGB orthomosaic overlaid on the 3D digital surface model (11 cm), with labeled example locations of the four primary target classes of interest. (**b**). Field photo (taken from photo icon on (**a**)) of emergent cattails (top half of photo), floating *S. polyrhiza* (*Duckweed* spp., yellowish-green vegetation), floating *N. odorata* (large lilies) and floating EFB (darker green, small lilies sparsely distributed in *S. polyrhiza* mat).

During each flight campaign, we collected ground control points (GCPs) and ground reference points using real-time-kinematic (RTK) GPS equipment (Emlid Reach™ and Rover™, Hong Kong, China). The purpose of the high-accuracy GPS data was to (1) georeference each UAS dataset using the GCPs [28], (2) use the ground reference data to train and test classification algorithms, and (3) interpret classification outputs using the community composition data. We collected the ground reference points based on expected differences in spectral signatures of the features between wetland cover classes on the given day of flight. We assigned the following wetland cover classes: Open Water (n = 4), Terrestrial Vegetation (n = 2), 'Emergent Vegetation' (dominated by *Typha* × *glauca*; n = 4), 'Submergent Vegetation' (*Chara* spp., *Ceratophyllum demersum*, *Elodea canadensis*,

*Myriophyllum* spp., or *Potamogeton crispus*; n = 5), Floating 'EFB and Nymphaea (Lily) spp.', and Floating 'Duckweed spp' (n = 4). All ground reference points were randomly assigned throughout the study site (depending on ground access) to achieve sufficient replication of each wetland cover class. At each ground reference point, we measured percent areal cover of the all plant species occurring within a ~30 cm$^2$ area around the GPS unit and assigned a vegetation class based on the dominant species present (>40% areal cover). Lastly, we corrected and processed the RTK GPS points using EZsurv™ (Effigis™, Montreal, QC, Canada) software, a post-processing software for correcting floating-point (RTK) coordinates using global navigation satellite-systems (GNSS) observation files from the RTK system. We created point shapefiles from the resulting .csv files of ground reference points and GCPs in R.

### 2.2. Image Processing

For absolute spatial accuracy, we imported and projected the GCPs into Pix4Dmapper (Pix4D version 4.5.6) prior to image and point cloud generation. For quality and filtering, we used a point cloud densification of $\frac{1}{4}$-image size with a minimum number of three matches. The 3D textured mesh was generated at default medium resolution with a matching window of 7 × 7 pixels. We then extracted the geotag information of each raw image and leveraged SfM photogrammetry (i.e., identifying common features in conjugate images as tiepoints) [36] to create an orthomosaic of the site for each spectral band at both flight altitudes at the automatic (i.e., user-specified) resolution. All spectral bands in both datasets were radiometrically calibrated to reflectance with the 'Camera and Sun Irradiance' correction in Pix4D to normalize potentially different illumination conditions between flights, based on corrections from an upward-facing irradiance sensor. In addition to the multispectral orthomosaic, we generated a Normalized Difference Vegetation Index (NDVI) raster from the NIR and Red bands, a Digital Surface Model raster from SfM point cloud (DSM; 2.9 cm & 11 cm; or modeled representation of the elevation of all features on the surface), and a Digital Terrain Model (DTM; 55 cm; or a modeled representation of the bare earth elevation) of the flight footprint. The latter two raster layers generated structural metrics (i.e., surface and terrain elevation in meters) and were hypothesized to supplement the spectral reflectance layers in the classification process. The average ground sampling distance (GSD) for each pixel in the high-resolution imagery was 10.99 cm (hereafter, 11 cm), covering an area of 0.101 km$^2$ (10.05 ha). For the very-high resolution data, the average GSD was 2.99 cm (hereafter, 3 cm) for each pixel, covering an area of 0.047 km$^2$ (4.73 ha). We set the project coordinate system of both datasets to WGS 1984, UTM Zone 17N, and both achieved georeferencing accuracy of mean RMSE values $\leq 0.001$ m for the X, Y, and Z coordinates. In total, the processing produced seven layers of data (four spectral bands, NDVI, surface and terrain models) × 2 for both flight altitudes.

### 2.3. Image Subsetting

Before stacking all raster layers into a composite dataset, we subsetted and aligned all grid cells (i.e., pixels) of the layers for each spatial resolution separately. We subsetted each dataset using *.exif* data on height of captured image and geolocation tags to calculate the width of each image footprint (via the *exifr* package in R) [37]. We then used a distance of seven meters to create a buffered polygon of all the GPS point locations of each photo and used the convex hull setting in the Minimum Bounding tool in ArcPy [38]. The resulting polygon indicates the area of the flight footprint with substantive image overlap and thereby excludes distorted margins of orthomosaics [38].

### 2.4. CHM vs. DSM for Vegetation Structure

Prior to creating image composites, we generated a Canopy Height Model (CHM) by subtracting the DTM from the DSM in each dataset. In doing so, we effectively removed non-vegetated terrain and the resulting CHM consists of vegetation height values only. However, because the DTM is of coarser resolution than the DSM, we resampled pixels

(via nearest neighbor algorithm) of the higher resolution DSM to that of the coarser DTM. Pixel aggregation avoids false accuracies of the pixel values, and thus we subsequently aggregated the remaining bands of the dataset to achieve coarser spatial resolution of 55 cm pixels for the high-resolution imagery and 15 cm pixels for the very-high resolution imagery. We aggregated pixels and created CHMs in R, as well as on-the-fly while creating composites in ArcGIS Pro 2.6.0.

We observed stark differences in canopy height across the wetland (e.g., Floating vs. Emergent vegetation; Figure 2), so we sought to utilize the generated CHMs to inform classification algorithms in distinguishing different vegetation functional types. Photogrammetric CHMs, along with multispectral information, have been useful in classifying vegetation in other systems [7,8,26,27]. However, given that we resampled all raster layers to the coarser resolution of the CHMs, this poses a tradeoff of obtaining canopy information vs. losing spatial resolution that may be key for distinguishing stands of different vegetation types. As such, we compared classification outputs from multispectral + CHM datasets for both resampled resolutions (15 cm for very-high resolution and 55 cm for high resolution) to those derived from the multispectral + surface-based structural metrics (DSM) that maintained the original, higher spatial resolutions (3 cm and 11 cm). Further, we calculated the standard deviation of a pixel's neighborhood (8 surrounding pixels) modeled surface or canopy height in a moving window [21] using the Focal Statistics tool in ArcGIS Pro 2.6.0. This textural metric, also known as surface or canopy rugosity (hereafter, "rugosity") [39,40] captures local variation in vegetation structure within and around pixels, and ought to be informative for classifying different types of invasive vegetation in our study site. The final multispectral + structural/textural stacks consisted of the four spectral bands, NDVI, DSM or CHM, and rugosity (of either surface or canopy height) at each spatial resolution. We also created multispectral-only stacks (spectral bands + NDVI) to compare classification accuracies with those from the multispectral + structural datasets (Tables 1 and 2).

**Table 1.** The spectral and structural bands generated from Pix4d for the classification analyses.

| Parrot Sequoia/pix4d-Generated Bands | Spectral Band Width (nm)/Structural/Textural Metric | Spatial Resolution (cm) | Vertical Range (m) |
|---|---|---|---|
| Green | 530–570 nm | 3, 11 | NA |
| Red | 640–680 nm | 3, 11 | NA |
| Red Edge | 730–740 nm | 3, 11 | NA |
| NIR | 770–810 nm | 3, 11 | NA |
| NDVI (included in Multispectral category below) | Index between 0–1 | 3, 11 | NA |
| Digital Surface Model | Surface Height in meters | 3, 11 | 178.50–192.30 (3 cm dataset) 177.97–194.21 (11 cm dataset) |
| Digital Surface Model Rugosity | SD of Surface Height of 8 neighbor pixels | 3, 11 | NA |
| Canopy Height Model | Canopy Height in meters | 15, 55 | −1.15–11.90 (15 cm dataset) −2.33–24.30 (55 cm dataset) |
| Canopy Height Model Rugosity | SD of Canopy Height of 8 neighbor pixels | 11, 55 | NA |

**Table 2.** The combinations of spectral, structural, and textural bands in the three raster composites (Multispectral Only, Multispectral + CHM metrics, Multispectral + DSM metrics) used as predictor variables to inform classification models.

| Classification Approach | Pixel Resolution | Dataset (with Pixel Resolution in cm) |
|---|---|---|
| Pixel-Based | High-Resolution | Multispectral Only (11 cm)<br>Multispectral + CHM + Rugosity (55 cm)<br>Multispectral + DSM + Rugosity (11 cm) |
| Pixel-Based | Very-high Resolution | Multispectral Only (3 cm)<br>Multispectral + CHM + Rugosity (15 cm)<br>Multispectral + DSM + Rugosity (3 cm) |
| Object-Based | High-Resolution | Multispectral Only (11 cm)<br>Multispectral + CHM + Rugosity (55 cm)<br>Multispectral + DSM + Rugosity (11 cm) |
| Object-Based | Very-high Resolution | Multispectral Only (3 cm)<br>Multispectral + CHM + Rugosity (15 cm)<br>Multispectral + DSM + Rugosity (3 cm) |

*2.5. Preliminary Classification, Image Masking, & Layer Stacking*

Our goal was to extract features from remotely sensed UAS data to classify and distinguish different functional types of wetland vegetation (Emergent, Submergent, and Floating classes) that were dominated by various invasive species, using the two common workflows. Prior to classification, we masked out pixels across the imagery that were not appropriate for this objective (i.e., Open Water and Terrestrial Vegetation). To mask out these areas, we uploaded the multispectral UAS imagery (both resolutions) into Google Earth Engine (GEE). We used a combination of ground reference points and on-the-ground familiarity of the site to create points of three different feature classes: "Terrestrial", "Wetland Vegetation", and "Open Water". We used stratified random sampling for each of these classes to achieve a suitable sample size of reference data (at least 50 points/class) [41]. We then implemented the *random forest* classifier in GEE to classify the entire image footprint and validated the classification outputs with hold-out points. Accuracies for all three classes were desirable (>95%), so we exported the classified raster to mask all multispectral and structural data. Prior to stacking all layers, we first clipped all layers from both data by the generated bounding polygons for each resolution (see Image Subsetting). Next, we overlaid 'Wetland Vegetation' pixels from the preliminary classified rasters and masked out undesired areas (i.e., Open Water and Terrestrial Vegetation) for both datasets. Once all layers consisted of wetland vegetation area only, we used the 'Composite Bands' raster function to stack all datasets into multiband rasters for each resolution in ArcGIS Pro 2.6.0.

*2.6. Reference & Data Extraction*

The field-collected ground reference dataset had an inadequate number of ground reference points due to a loss of connection in the RTK equipment while in the field. There were 13 ground reference points consisting of Submergent (n = 5), Floating (n = 4), and Emergent Vegetation (n = 4), below the generally acceptable sample size of at least 50 points per class [41]. To boost the class sample sizes of our ground reference data, we generated at least 200 points randomly distributed across the entire extent of each masked dataset. We then used the RGB orthomosaics to confirm each point as one of the three vegetation classes via image interpretation. After combining the digitized reference data with the field-collected ground reference data, we extracted pixel values of all raster bands to the intersecting point locations (for both datasets) to serve as predictor variables for further classification. The final sample size was greater than 50 points per class in the 11 cm dataset (N = 249) and for the 3 cm dataset (N = 229, Table 3). We then used the respective RGB orthomosaics to confirm each point as one of the three primary vegetation classes (while also distinguishing the two floating sub-classes). To verify that we classified our digitized

reference data with reasonable accuracy, we classified our 13 original ground reference points, based on location only, overlaid on the RGB imagery and compared to the field notes. For the points for which we had ground data, vegetation classes determined from RGB imagery agreed 80% on average with ground data (Floating 86%, Emergent 80%, Submergent 71%). To ensure as accurate of sampling as possible, we used the location of these points on the RGB and multispectral imagery to train image interpreters when boosting sample size of each dataset via image interpretation and on-screen digitizing.

**Table 3.** The number of field-collected, digitized from image interpreation, and total reference points (and intersecting pixels) for each vegetation class.

| Vegetation Class | Field Collected | Collected from Image Interpretation | Total Reference Points for Classification | Dataset |
|---|---|---|---|---|
| Emergent | 4 | 79 | 83 | 11 cm (55 cm for CHM models) |
| Floating | 4 | 96 | 100 | 11 cm (55 cm for CHM models) |
| Submergent | 5 | 61 | 66 | 11 cm (55 cm for CHM models) |
| Total | 13 | 236 | 249 | |
| Emergent | 4 | 79 | 83 | 3 cm (15 cm for CHM models) |
| Floating | 4 | 90 | 94 | 3 cm (15 cm for CHM models) |
| Submergent | 5 | 49 | 53 | 3 cm (15 cm for CHM models) |
| Total | 13 | 216 | 229 | |

## 2.7. Pixel-Based Classification Approach

We used the train function in the caret package [42] in R to run a *random forest* (RF) algorithm on our various datasets and classification approaches. We used 'Class' as the outcome variable and the five or seven bands (values extracted from pixels) as individual predictors with no interactions. We configured the parameters in this ensemble classifier to have at least five predictor variables (and up to 14) at each split (*mtry*) with a minimum terminal node size of five and 300 decision trees (*ntree*) in each RF run. To obtain average overall accuracy, we used repeated k-fold cross validation (k = 5) to split 4/5 of the dataset to train the RF model and test model predictions with the remaining 1/5 until all five folds were used to validate the model predictions. We ran models 50 times on these random-stratified splits of the data to obtain average accuracies. The total number of sample points used to validation predictions was 12,400 for the 11 cm data and 11,450 for the 3 cm data after the 50 model runs. We then created spectral profiles of the reference data (Figures S1–S5) as well as error matrices and mapped predictions from the best performing RF models.

## 2.8. Object-Based Classification Approach

We sought to compare pixel- and object-based approaches to optimize accuracy when classifying vegetation classes (or individual species) from UAS imagery [6]. For these datasets, vegetation stands of various sizes and individuals for each of the three functional types were usually greater in area than the 3 or 11 cm pixel sizes in our UAS data. We implemented the ArcGIS Mean Shift Segmentation algorithm through the *SegOptim* package in R, a newer approach that closely integrates optimization, segmentation and classification algorithms [43]. To optimize the three segmentation parameters in this function ('Spectral Detail', 'Spatial Detail', and 'Minimum Segment Size'), we ran a genetic algorithm model to obtain the best "fitness" values, or those parameters that yielded highest classification accuracies from subsequent RF model runs of 200–500 image segmentations of the datasets (depending on spatial resolution and time constraints; see Figure 3 for optimal outputs for each spatial resolution). To obtain these values, we set value ranges for each segmentation parameter that we thought to be reasonable in representing the realistic conditions of spectral and spatial ranges, along with minimum segment size, in the imagery. The genetic algorithm then randomly drew a value from these ranges for each individual segmentation. We set these values between 10–20 for Spatial and Spectral detail given the heterogeneity of features and spectral similarity of features/pixels (the higher the value, the more complex

spectral and spatial configuration of features in imagery, out of 20). We set Minimum pixel size from 2–50, depending on spatial resolution. We then used the optimal parameters for a final segmentation and RF classification, using the mean and SD of the band values (of each generated segment and the intersecting reference point) as predictor variables for the three vegetation classes.

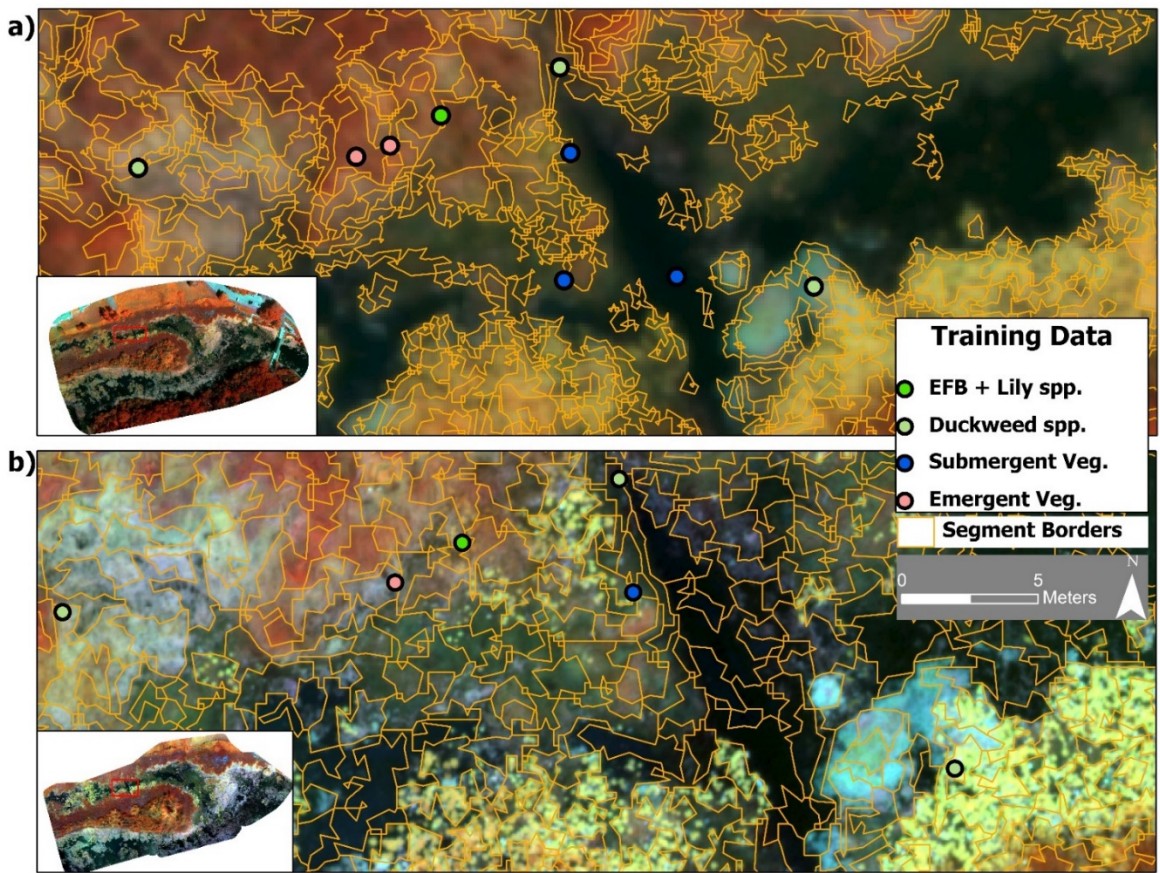

**Figure 3.** (**a**) The optimal segmentation layer for the 11 cm dataset (false color composite) with overlaid training data points from the four vegetation classes. (**b**) The optimal segmentation layer for the 3 cm imagery of the same area and training data points.

### 2.9. Accuracy Assessment and Model Performance

We utilized multiple accuracy metrics for evaluating the performance of the RF classifiers from both approaches. We first ran preliminary models on 'disaggregated' data among the Floating class, with floating Duckweed spp. and floating EFB + Lily spp. as separate classes due to their expected differences in spectral reflectance (Figure S1), as well as both species' prevalence in the wetland. However, we found lower accuracies with these classes separated than when we combined both floating classes into a single Floating class (Table S1 vs. Table 4). We report the full results of this analysis in the Supplementary Information. After that, we decided to proceed with models that predicted Floating (with the two subclasses combined) along with Emergent and Submergent vegetation. We justify this aggregation of vegetation classes due to the relative dominance of key invasive species in each class (e.g., Duckweed spp. dominance in the Floating class, EFB and *Typha* × *glauca* dominating the Emergent class) observed from independent field data at Alpena [44].

**Table 4.** The performing classification outputs for three different combinations of data for each approach at both spatial resolutions. Overall accuracies (in %) for each datasets of the three main functional types of wetland vegetation in the high-resolution UAS imagery (11 cm) at Alpena Wildlife Refuge. The best results for each spatial resolution are highlighted in bold; PA is Producer Accuracy (100%-omission error) and UA is User Accuracy (100%-commission error). Mean AUROC and F1 is the mean metric of all three classes.

| Classification Approach | Spatial Resolution | Data | OA | Floating | | Emergent | | Submergent | | Mean AUROC | Mean F1 Score |
|---|---|---|---|---|---|---|---|---|---|---|---|
| | | | | PA | UA | PA | UA | PA | UA | | |
| *Pixel-based* | 11 cm | Multispectral + CHM + Rugosity (resampled to 55 cm) | 76.9 (95% CI 0.761–0.769) | 81.8 | 75.7 | 88.9 | 78.7 | 48.1 | 75.7 | 86.0 | 78.6 |
| *Object-based* | 11 cm | Multispectral + CHM + Rugosity (resampled to 55 cm) | 73.4 (95% CI 0.725–0.743) | 83.7 | 70.0 | 84.8 | 80.5 | 34.6 | 65.1 | 82.4 | 76.3 |
| *Pixel-based* | 11 cm | Multispectral + DSM + Rugosity | **81.4 (95% CI 0.807–0.821)** | **86.3** | 76.7 | **90.0** | 84.6 | 63.3 | **86.7** | **90.0** | 80.6 |
| *Object-based* | 11 cm | Multispectral + DSM + Rugosity | 77 (95% CI 0.762–0.776) | 81.9 | 72.9 | 86.6 | 84.3 | 54.6 | 73.0 | 86.9 | 77.1 |
| *Pixel-based* | 11 cm | Multispectral Only | 80.8 (95% CI 0.801–0.815) | 84.9 | **76.8** | 89.6 | **85.7** | **63.6** | 81.3 | 87.6 | **80.7** |
| *Object-based* | 11 cm | Multispectral Only | 77.3 (95% CI 0.765–0.78) | 83.2 | 73.1 | 88.9 | 83.5 | 50.1 | 75.2 | 86.8 | 77.8 |
| *Pixel-based* | 3 cm | Multispectral + CHM + Rugosity (resampled to 15 cm) | 58.3 (95% CI 0.574–0.592) | 57.2 | 53.2 | 79.5 | 73.2 | 29.4 | 38.7 | 74.8 | 55.1 |
| *Object-based* | 3 cm | Multispectral + CHM + Rugosity (resampled to 15 cm) | 76.7 (95% CI 0.759–0.774) | 71.8 | 70.4 | 83.1 | 82.9 | **75.0** | 77.8 | 88.8 | 71.1 |
| *Pixel-based* | 3 cm | Multispectral + DSM + Rugosity | **78.9 (95% CI 0.774–0.790)** | **74.3** | 73.8 | 86.5 | **82.8** | 72 | 78.4 | 74.8 | **74.1** |
| *Object-based* | 3 cm | Multispectral + DSM + Rugosity | 77 (95% CI 0.763–0.778) | 72.0 | 72.9 | 86.6 | 81.5 | 70.8 | 76.9 | **89.5** | 72.5 |
| *Pixel-based* | 3 cm | Multispectral Only | 72.6 (95% CI 0.717–0.734) | 66.3 | 68.3 | 82.9 | 76.1 | 67.6 | 74.2 | 85.8 | 67.3 |
| *Object-based* | 3 cm | Multispectral Only | 76.6 (95% CI 0.758–0.773) | 70.7 | **74.2** | **86.9** | 77.7 | 70.5 | **78.7** | 87.6 | 72.4 |

For all models, we calculated overall accuracy (OA), user's accuracy (UA; 1 − % commission error), and producer's accuracy (PA; 1 − % omission error). In addition, we assessed how well the classifier ranked patterns by calculating the Area Under the Receiver Operating Characteristic curve (AUROC) for each class (vs. the other two aggregated classes). We also report F1 scores (harmonic average of UA and PA) of each model as useful metrics to account for the imbalance of class sample sizes [45]. For both AUROC and F1 scores, we report the mean metric from the three classes. We also overlaid and sampled the classified areas onto Mohammadi et al.'s [44] quadrat points and compared relative abundance of each species, as well as non-native and/or invasive status, within and across the classified areas to those classified in the field (see more details of field data collection in Supplementary Materias). Finally, we plotted predictions of best performing models across the multispectral rasters for both spatial resolutions in R.

## 3. Results

### 3.1. Very-High (3 cm) vs. High-Resolution (11 cm) Pixels

The high-resolution data yielded the highest overall accuracies (OA), and most other metrics, compared to the very-high resolution data (Table 4). Within each dataset, accuracies yielded only slight differences across the six models, but the 'Multispectral + DSM + Rugosity' dataset resulted in the best performing model at both spatial resolutions. At the 11 cm spatial resolution, the 'Multispectral + DSM + Rugosity' model produced the highest PA for Floating and Emergent classes, the highest UA for Submergent Vegetation, as well as the highest Mean AUROC. The 11 cm 'Multispectral Only' model produced the highest UA for Floating and Emergent vegetation, as well as the highest Mean F1 Score. At the 3 cm spatial resolution, models using "Multispectral + DSM + Rugosity' produced the highest PA for Floating, highest UA for Emergent, and highest Mean AUROC and F1 scores. The 'Multispectral + CHM + Rugosity' model produced the highest PA for Submergent vegetation, while 'Multispectral Only' models produced the highest UA for Floating and Submergent vegetation, and highest PA for Emergent vegetation.

Emergent vegetation had the highest PA and UA among classes, regardless of spatial resolution, while models generally yielded the poorest PA/UA for Subgmerent vegetation in both datasets (Table 4). For the overall top performing model, the 11 cm 'Multispectral + DSM + Rugosity,' the most confusion for the classification occurred between the Floating and Emergent classes (Table 5). Predicted Submergent points were more often misclassified as Floating than as Emergent (Table 5). When predictions were applied to the entire 11 cm imagery footprint, the greatest number of pixels were assigned to the Floating class with the least number of pixels assigned to the Submergent class (Figure 4).

**Table 5.** Confusion matrix for the best performing model (11 cm 'Multispectral + DSM + Rugosity'). Reference data are actual recorded classes from the field (including reference points determined from the imagery) and predicted points were generated from the *random forest* model. The greyed out diagonal values are the number of points correctly classified by the model, with the bold value in the bottom right cell representing the overall accuracy of the model (sum of the diagonal values/the total number of points × 100). Producer's and User's Accuracy (PA & UA) are displayed in % accuracy.

| | | Reference | | | | |
|---|---|---|---|---|---|---|
| | | Floating | Submergent | Emergent | Total | UA |
| | Floating | 4317 | 947 | 366 | 5630 | 76.68 |
| | Submergent | 276 | 2090 | 45 | 2411 | 86.69 |
| **Predicted** | Emergent | 407 | 263 | 3689 | 4359 | 84.63 |
| | Total | 5000 | 3300 | 4100 | 12,400 | |
| | PA | 86.34 | 86.34 | 86.34 | | **81.42** |

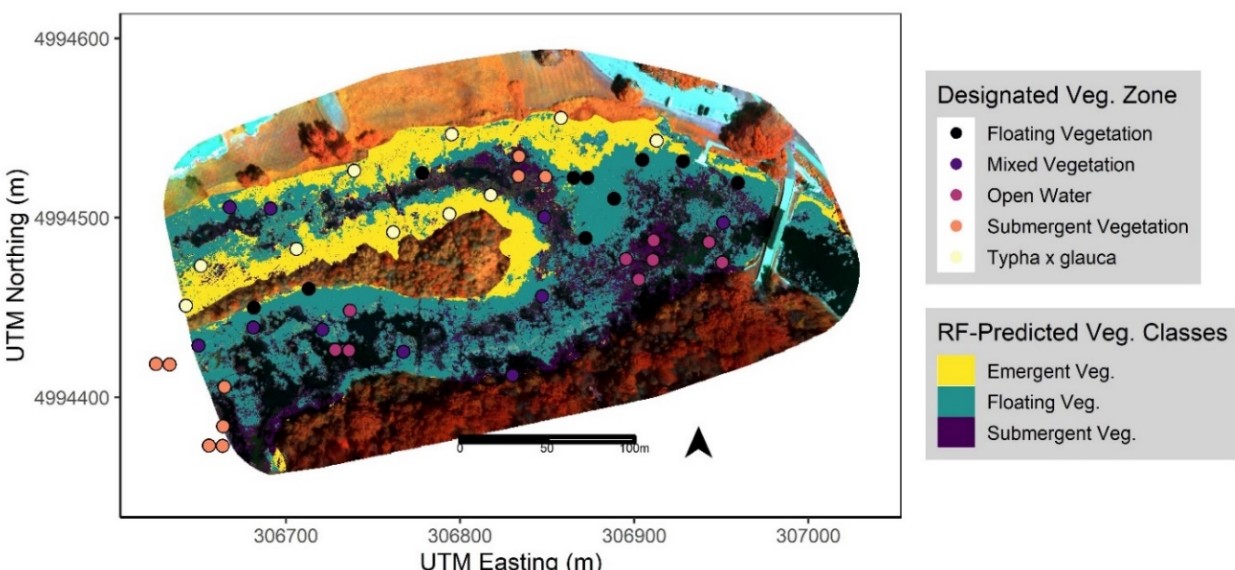

**Figure 4.** Mapped predictions for each wetland vegetation class from the best performing RF model ('Multispectral + DSM + Rugosity' 11 cm data via a pixel-based approach) with overlaid field data points of vegetation zones determined by Mohammadi et al. (*in prep*). Model predictions are mapped across the extent of wetland vegetation only, with unmapped areas including masked-out terrestrial and open water features for reference.

### 3.2. Pixel- vs. Object-Based Classification Approach

A pixel-based approach yielded the highest accuracies in the high-resolution data as well as the top model in the very-high resolution data (Table 4). Across all models in each dataset, pixel-based models outperformed object-based models. However, within the very-high resolution data, all three object-based models yielded OAs within 2.3 percentage points of the top performing pixel-based model. For 11 cm models, pixel-based models outperformed object-based models by up to 8 percentage points in OA.

### 3.3. Multispectral Only vs. Multispectral plus Structural Layers

Similar to the classification approach, the inclusion of DSM + Rugosity along with multispectral data resulted in the best performing models in both pixel- and object-based classification at both high and very-high resolutions. However, models including CHM did not perform as well as the more parsimonious 'Multispectral Only' models at high resolution (Table 4). Specifically, including DSM metrics in the 11 cm dataset yielded higher accuracies than the datasets with the CHM metrics, with a similar trend occurring in the very-high resolution data. Notably, the pixel-based 3 cm dataset with CHM metrics performed substantially worse than the five other models in the very-high resolution data. Lastly, the highest mean AUROC and F1 scores often included both spectral and structural (DSM) data, with the exception of the 11 cm 'Multispectral Only' yielding the highest mean F1 score overall.

### 3.4. Predicted Vegetation Classes Sampled to Quadrat Data

After extracting the predicted classes from the best performing RF model to the intersecting quadrat points (Figure 4), we calculated the mean relative abundance (proportion of total plant cover) of the three most common species recorded within each class. We observed inconsistent patterns of each species recorded at the points in the field vs. the primary functional group defined by the RF classifier based on the UAS data. Namely, we observed that EFB (*H. morsus-ranae*) was present and/or dominant across all three vegetation classes, submergent species (*C. demersum*) present in areas classified as Floating, and some floating species (*N. odorata*, *H. morsus-ranae*) present in areas classified as Submergent vegetation (Table 6). We also found that two of three dominant species in the Emergent

class, the most accurately predicted class overall (Table 4), were considered non-native and/or invasive to the study site (Table 6). Lastly, points classified as 'Open Water' in the field that overlaid on masked-out areas, from the preliminary classification, were classified as such (Figure 4).

**Table 6.** Relative abundance (proportion of total plant cover) of the three most common species within each predicted vegetation class. The designated vegetation zone indicates what each of the field point was designated as, per Mohammadi et al. (*in prep*). The % agreement in the first column denotes the proportion of field-designated points that were classified as the same by the RF classifier predictions. "EFB and Cattail present?" show how many of the field points recorded either species out of the total assigned to that predicted class. The "Non-native and/or invasive?" indicates whether the three most common species within each class are considered non-native and/or invasive at the study site.

| Designated Vegetation Zone (Mohammadi et al. *in prep*) | RF Predicted Vegetation Class (Total Number of Points) * | Species | Relative Abundance (% Cover/Total Cover) | Percent Cover (%) | EFB Present? | Cattail Present? | Non-Native and/or Invasive? (Yes or No) |
|---|---|---|---|---|---|---|---|
| Emergent (90% agreement) | Emergent (n = 11) | *Hydrocharis morsus-ranae* | 33.7 | 28.9 | 100% of points | 90.9% of points | Y |
| | | *Spirodela polyrhiza* | 24.4 | 21.0 | | | N |
| | | *Typha × glauca* | 21.7 | 18.6 | | | Y |
| Floating (90% agreement) | Floating (n = 12) | *Nymphaea odorata* | 14.4 | 14.3 | 91.6% of points | 16.6% of points | N |
| | | *Ceratophyllum demersum* | 10.4 | 10.3 | | | N |
| | | *Hydrocharis morsus-ranae* | 7.8 | 7.8 | | | Y |
| Submergent (33% agreement) | Submergent (n = 3) | *Nymphaea odorata* | 72.2 | 4.3 | 33.3% of points | 0% of points | N |
| | | *Spirodela polyrhiza* | 16.7 | 1.0 | | | N |
| | | *Hydrocharis morsus-ranae* | 5.6 | 0.03 | | | Y |
| Open Water (47% agreement) | Open Water (n = 22) ** | *Ceratophyllum demersum* | 22.4 | 1.0 | 31.8% of points | 4.5% of points | N |
| | | *Spirodela polyrhiza* | 14.3 | 5.0 | | | N |
| | | *Nuphar variagatum* | 12.4 | 4.3 | | | N |

* Two of the 50 total field points (see more info in SI; Mohammadi et al. *in prep*) were located outside of the flight footprint, and thus were not designated with any of the RF- predicted vegetation classes. ** Field points predicted as "Open Water" were located in masked-out areas from our analysis (see methods) but still within the boundaries of the UAS flight footprint and wetland vegetation, thus they were designated as "Open Water" via visual interpretation.

## 4. Discussion

This study assessed our ability to map wetland vegetation functional types that are dominated by invasive plants (EFB and cattails) using UAS imagery. Specifically, we compared how classification accuracy changed when using (a) high- vs. very-high resolution UAS data, (b) implementing a pixel- vs. object-based classification approach, and (c) using only multispectral data vs. multispectral plus structural data. Surprisingly, we obtained higher classification accuracies from the high-resolution data than the very-high resolution data with no clear trend for accuracy between pixel- and object-based approaches, but slightly higher accuracy when we incorporated structural data despite the trade-off in spatial resolution. Overall, there is potential to utilize our workflow to detect wetland vegetation functional types that include invasive species using UAS data. Below we discuss specific results, limitations, and potential management implications within each objective.

### 4.1. Very-High (3 cm) vs. High-Resolution (11 cm) Pixels

The most surprising finding from our study was that the very-high resolution yielded lower accuracies than the high-resolution data. We expected that the very-high resolution data of 3 cm pixels would better capture key morphometric differences between each vegetation class. However, the heterogeneity of features and biophysical characteristics of species within each class likely played a role in determining which pixel size was best for accurate predictions. For example, mature EFB, the dominant invasive plant in the Emergent and Floating classes, have leaves that are 5–6 cm in diameter. Therefore, we would expect that 3 cm pixels would be able to delineate individual EFB leaves. Yet, this resolution may have captured other fine-scale features interspersed with EFB, such as cattail leaf litter or small water gaps in between floating leaves [6,24], obscuring the RF's ability to delineate this species accurately. In 11 cm pixels, much of this heterogeneity is merged into mixed, but evidently informative pixels that delineate key differences in the three functional groups as exhibited by the lower standard error in many spectral and structural bands for each class (Figures S1–S5). We emphasize that this was only the case for the Emergent and Floating classes, whereas we achieved higher producer accuracy for Submergent vegetation at the very-high resolution. One explanation for this trend is that the smaller 3 cm pixels, along with distinct spectral and structural values for this class (Figure S3), better captured the smaller, interspersed areas of Submergent vegetation in the masked image footprint. Previous research in UAS forestry classification also found higher classification accuracies in imagery of coarser spatial resolution depending on the density of tree stands [46]. Practitioners should carefully consider the biophysical characteristics and spatial patterns of the target species (or functional group) when selecting the appropriate spatial resolution to fly UAS for detecting invasive vegetation.

### 4.2. Pixel- vs. Object-Based Classification Approach

Our results indicate that a pixel-based approach performed slightly better than object-based methods using the high-resolution data, although this pattern was not as obvious in the very-high resolution data. Previous UAS studies found that object-based methods were generally more effective than pixel-based methods at classifying invaded stands in terrestrial ecosystems [5,6]. We attribute the difference in our findings to the mixed distribution of species across the functional classes that may have prevented the segmentation algorithms from creating image objects with relatively homogenous spectral or structural values, or in area. Specifically, one species in the Floating class, *S. polyrhiza*, which is spectrally and structurally similar to EFB, was prevalent throughout the site, growing in between mature EFB and Emergent vegetation. Similarly, Abeysinghe et al. 2019 [7] found that another abundant duckweed species, *L. minor*, caused object confusion for their classifier due to obscured object edges between vegetation classes in UAS imagery of another Great Lakes wetland site. Dominant EFB within and around Emergent stands may also have created objects with a mix of species and in slightly coarser 11 cm pixels,

as observed with the confusion between classifying Floating and Emergent vegetation in our best performing model. The mixed spatial distribution of species across vegetation zones resulted in less distinct spectral and structural bands of the image objects for the classes, and likely led to lower classification accuracy [47]. In the 3 cm data, the finer spatial resolution better captured these species, producing more appropriately sized objects with distinct spectral and structural values compared to the segmentation outputs of the 11 cm data (Table S2, Figure S5). Despite achieving optimal segmentation parameters, we suggest for future studies to consider the complex spatial distributions of wetland vegetation when determining whether a pixel- vs. object-based approach is best for classifying invasive plants in UAS imagery.

### 4.3. Multispectral Only vs. Multispectral plus Structural Layers

Including structural data (namely, DSM-derived values) with the multispectral bands generally improved classification accuracy over using multispectral data only. In past UAS studies, researchers have elucidated the benefits of CHMs that reconstruct 3D structure of image features as a supplement to the spectral data for mapping heterogeneous vegetation [7,8,23,26,27,48]. Across spatial resolutions, our results tend to show that incorporating structural data does increase accuracy, although we observed that DSM structural and textural layers generally achieved higher accuracy than CHM layers. In the high-resolution data, the DSM layers, and even 'Multispectral Only' data, outperformed the CHM data, whereas DSM data provided only slightly higher accuracies in the very-high resolution data. The coarser resampled resolution of the high-resolution CHM data (55 cm pixels) may have mixed important spectral and structural features that distinguish the three vegetation classes at our study site. The resampled resolution (15 cm pixels) in the very-high resolution CHM data, via an object-based approach, may have been more informative for maintaining these distinctive features. Moreover, there may have been errors within the DTM layers, as we observed negative values in both CHM layers at both resolutions, obscuring true heights recorded in pixels or objects of wetland vegetation. The attenuation of light in the water column could have contributed to inaccurate reconstruction of the 'bare terrain' at our site, as found in imagery of wetland plant communities from other remote sensing platforms [49]. To compensate for these errors, we agree with the recommendation by Martin et al. (2018) [8] to instead fly UAS multiple times over a site throughout the year and incorporate other 2D textural data, such as grey level co-occurrence matrices, for improving mapping accuracy of invasive vegetation. To our knowledge, our results are the first to show the utility of structural data derived from DSMs can be just as or more beneficial than structural data from CHMs when classifying invasive wetland vegetation in UAS imagery.

### 4.4. Importance of UAS Flight Parameterization and Spectral Resolution of UAS Sensor

To improve future efforts of reconstructing 3D structural features of invasive wetland vegetation, it is important to assess how varying forward and side overlap of individual images, and adjusting flight speed, influence classification accuracy [28–30]. One reason for the lower accuracies from the very-high resolution data was perhaps due to suboptimal flight parameters for that flight altitude (85% front and side overlap of images flown at 30 m altitude) over our study site. Seifert et al. (2019) [29] found that higher image capture frequency (forward and side overlap > 90%) at lower flight altitudes (15–20 m above canopy) yielded highest precision for reconstructing details of tree crowns in 3D point clouds. Therefore, the 85% overlap may have not been adequate for the image processing software to find common points (also known as matched keypoints) between consecutive images at 3 cm pixel resolution. As pixel resolution of small-scale features become finer, image overlap should increase to optimize the number of matched keypoints between images for accurate 3D point clouds of reconstructed features in UAS imagery [29]. In our datasets, we found a 4% greater number of matched keypoints per image, resulting in 100% calibrated images, in the 11 cm orthomosaics and 3D models (high altitude flights) than the number of matched keypoints per image, resulting in only 93% of calibrated

images in the 3 cm data (low altitude flights). Therefore, this insufficient parametrization of the image overlap potentially affected classification accuracy of the very-high resolution datasets. Considering tradeoffs between varying these flight parameters, processing time and importance of spatial scales in reconstructing target plant communities in UAS imagery will ensure that researchers obtain accurate structural values for classification purposes.

It is important to note that there were clear differences in how well we were able to predict the three different wetland classes, partially due to the spectral resolution of our sensor. Notably, omission and commission errors were lowest for the Emergent class, regardless of spatial resolution. The Emergent vegetation class exhibited distinct mean values across most spectral and structural bands, which likely played a role in the RF models' ability to discern patterns and make clear decisions about classifying Emergent points vs. the other two classes. Across spatial resolutions, the most confusion occurred between the Submergent and Floating classes. The spectral resolution of our sensor may have been inadequate (i.e., inappropriate bandwidths and/or number of bands) to distinguish both classes with such similar structural values, as observed in previous RS studies of riparian and marine systems [6,47]. Future research should assess the potential benefits of using higher spectral resolution sensors (e.g., hyperspectral) on UAS in attempts to distinguish interspersed floating and submergent vegetation in wetlands.

## 5. Conclusions

Based on our results, we believe that our UAS classification workflow is operational for detecting and monitoring invasive wetland vegetation. We showed that our classification scheme is appropriate for mapping wetland functional types dominated by target invasive species (e.g., dominant EFB and cattails in the Emergent class). The range of accuracy metrics did not vary significantly across the six models and approaches within each spatial resolution, although the pixel-based approach most often produced the highest accuracies. Therefore, managers could save computational time by avoiding the optimization of segmentation parameters in object-based methods [15,16]. Moreover, since high-resolution data produced higher accuracies than very-high resolution data, end-users can feel more confident in predicted vegetation classifications across pixels in larger image footprints while avoiding longer flight times to obtain <5 cm pixels of the same area. Lastly, we believe that managers can achieve desirable results from flying UAS during peak growth periods by utilizing the additional structural data generated from photogrammetric processing of UAS imagery. Overall, combining UAS data with machine learning algorithms is a powerful new tool for the early detection and rapid response to invasive wetland vegetation.

**Supplementary Materials:** The following are available online at https://www.mdpi.com/article/10.3390/rs13234733/s1, Figure S1: Spectral (a), structural (d,e), and textural (b,c) profiles of the disaggregated vegetation classes: Emergent, Submergent, floating EFB + Lily spp., and floating Duckweed spp. (*S. polyrhiza*) extracted from mean pixel values in the 11cm dataset. All bands, except Rugosity and Surface Height (m), are % reflectance values calibrated for each band during pix4D processing, Figure S2: Spectral (a), textural (b,c), and structural (d,e) profiles of the three aggregated vegetation classes: Emergent, Submergent, Floating Vegetation, extracted from mean pixel values 11cm dataset. All bands, except Rugosity and Surface Height (m), are % reflectance values calibrated for each band during pix4D processing., Figure S3: Spectral and textural profiles of the three vegetation classes: Emergent, Submergent, Floating Vegetation, extracted from mean pixel values in the 3 cm dataset. All bands, except Rugosity and Surface Height (m), are % reflectance values calibrated for each band during pix4D processing, Figure S4: Spectral and textural profiles of the three aggregated vegetation classes: Emergent, Submergent, Floating Vegetation, extracted from mean object values in the 11cm dataset. All bands, except Rugosity and Surface/Canopy Height (m), are % reflectance values calibrated for each band during pix4D processing, Figure S5: Spectral and textural profiles of the three vegetation classes: Emergent, Submergent, Floating Vegetation, extracted from mean object values in the 3 cm dataset. All bands, except Rugosity and Surface Height (m), are % reflectance values calibrated for each band during pix4D processing, Figure S6: Mean Decrease in Gini Importance (Importance on x-axis) for each mean and SD reflectance/structure for the 7 bands

in the pixel-based "Multispectral + CHM' model of the 3 cm data (resampled to 15 cm). Importance is based on % increase in mean squared error (MSE) for each band on the out of bag data for each tree in the RF and then computed after permuting a variable (band),. Figure S7: Mean Decrease in Gini Importance (Importance on x-axis) for each mean and SD reflectance/structure for the 7 bands in the pixel-based "Multispectral + DSM' model of the 3 cm data. Importance is based on % increase in mean squared error (MSE) for each band on the out of bag data for each tree in the RF and then computed after permuting a variable (band),. Figure S8: Mean Decrease in Gini Importance (Importance on x-axis) for each mean and SD reflectance/structure for the five bands in the pixel-based "Multispectral Only' model of the 3 cm data. Importance is based on % increase in mean squared error (MSE) for each band on the out of bag data for each tree in the RF and then computed after permuting a variable (band),. Figure S9: Variable importance for each mean and SD reflectance/structure of the segments (image-objects) for the 14 bands in the object-based 'Multispectral + Both CHM' model of the 3 cm data (resampled to 15 cm). Importance is based on % increase in mean squared error for each band on the out of bag data for each tree in the RF and then computed after permuting a variable (band),. Figure S10: Variable importance for each mean and SD reflectance/structure of the segments (image-objects) for the 14 bands in the object-based 'Multispectral + Both DSM' model of the 3 cm data. Importance is based on % increase in mean squared error for each band on the out of bag data for each tree in the RF and then computed after permuting a variable (band),. Figure S11: Variable importance for each mean and SD reflectance/structure of the segments (image-objects) for the 10 bands in the object-based 'Multispectral + Only' model of the 3 cm data. Importance is based on % increase in mean squared error for each band on the out of bag data for each tree in the RF and then computed after permuting a variable (band),. Figure S12: Mean Decrease in Gini Importance (Importance on x-axis) for each mean and SD reflectance/structure for the 7 bands in the pixel-based "Multispectral + CHM' model of the 11 cm data (resampled to 55 cm). Importance is based on % increase in mean squared error (MSE) for each band on the out of bag data for each tree in the RF and then computed after permuting a variable (band),. Figure S13: Mean Decrease in Gini Importance (Importance on x-axis) for each mean and SD reflectance/structure for the 7 bands in the pixel-based "Multispectral + DSM' model of the 11 cm data. Importance is based on % increase in mean squared error (MSE) for each band on the out of bag data for each tree in the RF and then computed after permuting a variable (band),. Figure S14: Mean Decrease in Gini Importance (Importance on x axis) for each mean and SD reflectance/structure for the 7 bands in the pixel-based "Multispectral Only' model of the 11 cm data. Importance is based on % increase in mean squared error (MSE) for each band on the out of bag data for each tree in the RF and then computed after permuting a variable (band),. Figure S15: Mean Decrease in Gini Importance (Importance on x axis) for each mean and SD reflectance/structure of the segments (image-objects) for the 14 bands in object-based "Multispectral + CHM' model of the 11 cm data (resampled to 55 cm). Importance is based on % increase in mean squared error (MSE) for each band on the out of bag data for each tree in the RF and then computed after permuting a variable (band),. Figure S16: Mean Decrease in Gini Importance (Importance on x axis) for each of the 14 bands in the object-based 'Multispectral + DSM' model of the 11 cm data. Importance is based on % increase in mean squared error (MSE) for each band on the out of bag data for each tree in the RF and then computed after permuting a variable (band),. Figure S17: Mean Decrease in Gini Importance (Importance on x axis) for each of the 10 bands in the "Multispectral Only' dataset object-based approach for the 11 cm data. Importance is based on % increase in mean squared error (MSE) for each band on the out of bag data for each tree in the RF and then computed after permuting a variable (band),. Figure S18: (a) The optimal segmentation layer for the 11 cm dataset (false color composite) with overlaid training data points from the four vegetation classes. (b) The optimal segmentation layer for the 3 cm imagery of the same area and training data points. Red boxes represent areas with varying degrees of segmentation, where OS indicates "Over-Segmented," RS indicates "Realistically-Segmented," and US indicates "Under-segmented. Table S1: Confusion matrix for the 11 cm 'Multispectral + DSM + Rugosity' model on the disaggregated vegetation classes recorded in the field: floating EFB + Lily spp., floating Duckweed spp., Submergent, and Emergent vegetation. Reference data are actual recorded classes from the field as well as reference points determined from the imagery and predicted classes at test points were generated from the random forest model. The greyed out diagonal values are the number of points correctly classified by the model, with the bold value in the bottom right cell representing the overall accuracy of the model (sum of the diagonal values/the total number of points * 100). Producer's and User's Accuracy (PA & UA) are displayed in % accuracy., Table S2: Mean object size (±SD, in m2) of the points for the

four vegetation classes from the reference data overlaid on the optimal segmentation outputs for the OBIA.

**Author Contributions:** Conceptualization, L.W.J., J.B., A.M., S.L. and J.T.; Methodology, L.W.J., J.B., A.M., S.L., M.C. and N.K.; Software, L.W.J., A.M., J.T. and N.K.; Validation, L.W.J.; Formal Analysis, L.W.J., J.B. and S.L.; Investigation, L.W.J. and J.B.; Resources, N.K. and J.T.; Data Curation, L.W.J.; Writing—Original Draft Preparation, L.W.J.; Writing—Review & Editing, J.B., S.L., M.C., A.M. and N.K.; Visualization, L.W.J.; Supervision, L.W.J., J.B. and S.L.; Project Administration, S.L.; Funding Acquisition, J.B., S.L. and A.M.; All authors have read and agreed to the published version of the manuscript.

**Funding:** This research was funded by the Michigan Department of Natural Resources Grant IS17-2006.

**Institutional Review Board Statement:** Not applicable.

**Informed Consent Statement:** Not applicable.

**Data Availability Statement:** The data and scripts that support the findings of this study are available as repositories on the corresponding author's github profile (https://github.com/ljochems?tab=repositories, accessed 18 November 2021).

**Acknowledgments:** Special thanks to the Michigan Department of Natural Resources, Invasive Species Grant Program for funding this research. Thanks to R. Mohammadi for supplying her field data to supplement this research. Thanks to T. Buhl, M. O'Brien, and L. St. John for help sampling in the field. Lastly, thanks to the University of Michigan Biological Station for the summer research fellowships.

**Conflicts of Interest:** The authors declare no conflict of interest.

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
