# Peer review of "Comparison of Different Analytical Strategies for Classifying Invasive Wetland Vegetation in Imagery from Unpiloted Aerial Systems (UAS)"

_remotesensing, doi:10.3390/rs13234733_

Round 1

Reviewer 1 Report

I read the manuscript two time. Having been a reviewer for 15+ years, it has never happened to me that I had no comments. Very well developed manuscript with significant merit. Great job

Author Response

Authors’ response: Thank you very much for the positive comment!

Reviewer 2 Report

This research conducted a rigorous comparison of different methodologies for identifying and mapping invasive Emergent and Floating vegetation, as well as Submergent species using a machine learning based classifier of random forest. The authors compared the accuracies using different spatial resolutions and two classification methods, as well as structural with multispectral data versus multispectral data alone. The reviewer believes that the current version of the manuscript is not yet ready for publication; the authors are encouraged to consider the following comments and suggestions and revise the manuscript accordingly.

  1. The authors should consider streamlining the Abstract section. Currently, the Abstract section is not in a natural flow and it provides a lot of information. The authors should make it more concise. The authors should avoid using acronyms in the Abstract section. The authors should also consider splitting the Introduction section into two sections, including an Introduction section and a Background (or Related Work) section. The introduction section should focus on introducing the research objectives and research questions, while the Background section should focus on literature review of related work and discussing why a review paper is important on this subject. The authors should also review more related literature. The authors should read and cite the paper of “the impact of small unmanned airborne platforms on passive optical remote sensing: a conceptual perspective”.
  2. The authors should use ground sampling distance (GSD) instead of spatial resolution. The authors still can use spatial resolution to indicate the specifications of the collected aerial photos but GSD is more appropriate for when discussing the size of the pixels.
  3. One of the fundamental questions that needs to be answered is that why pixel-based image analysis methods are discussed in the manuscript. These methods may not be suitable for high-spatial resolution aerial images.
  4. The RTK system used by the research team can collected ground control points information very well. However, the authors did not discuss how the collected coordinate information was post-processed. What software was used for this processing?
  5. The authors should discuss that the evaluation of the locational data should be conducted in terms of root mean squared error (RMSE), as specified by the American Society for Photogrammetry and Remote Sensing (ASPRS) and International Society for Photogrammetry and Remote Sensing (ISPRS).
  6. Most of the figures need to be improved. For example, in Figure 1-3, the reviewer has to zoom in at least 200% to be able to read. If at all possible, please create vector images for readability. In addition, the authors need to improve the tables.

Reviewer 3 Report

Please view the attached word document to access the reviewer revisions. 

Round 2

Reviewer 2 Report

The authors have addressed all my comments except for routine spelling/grammar check.

Author Response

Thank you for such helpful comments in improving our manuscript. We have done a final spelling/grammar check and will resubmit the final document.